# Free Methylglyoxal and Lactate Produced and Released by Cultured Cancer and Non-Cancer Cells: Implications for Tumor Growth and Development

**DOI:** 10.3390/cells14120931

**Published:** 2025-06-19

**Authors:** Dominique Belpomme, Philippe Irigaray, Jean-Marc Alberto, Clément Poletti, Charlotte Hinault-Boyer, Stéphanie Lacomme

**Affiliations:** 1Department of Cancer Clinical Research, Paris V University Hospital, 75015 Paris, France; 2European Cancer and Environment Research Institute, 1000 Brussels, Belgium; 3Association for Research on Treatment Against Cancer, 75015 Paris, France; 4Inserm UMRS 1256 NGERE—Nutrition, Genetics, and Environmental Risk Exposure, University of Lorraine, 54000 Nancy, France; 5Laboratoire Bioavenir, 57000 Metz, France; 6Université Nice Côte d’Azur, Centre Hospitalier Universitaire de Nice, Laboratoire d’Hormonologie, Hôpital Pasteur, 06204 Nice, France; 7Université Nice Côte d’Azur, Inserm U1065, C3M (Centre Méditerranéen de Médecine Moléculaire), Equipe 5 (Cancer, Métabolisme et Environnement), 06204 Nice, France; 8Centre de Ressources Biologiques, BB-0033-00035, CHRU, 54500 Nancy, France

**Keywords:** methylglyoxal, lactate, Warburg effect, glycolysis, tumor growth, cancer cells, stromal cells, cancer-associated fibroblasts, tumor microenvironment, cell metabolic reprogramming

## Abstract

We have previously shown that in cancer patients, free methylglyoxal (MG), a side-product of glycolysis, is recovered from tumors at significantly higher levels than from their corresponding non-cancerous tissues. We also recently confirmed our initial experimental finding that in these patients, free MG peripheral blood levels correlate positively with tumor growth, making free MG levels a new metabolic biomarker of tumor growth of interest to detect cancer and clinically follow cancer patients with no available biomarkers. Now we measure free MG and lactate produced by different cancer and normal cells cultured at low or high glucose concentration and in normoxic or hypoxic conditions to question whether cancer cells and non-cancer cells in tumors produce and release free MG and lactate. Surprisingly, we found that normal fibroblastic and endothelial cell lines grown in normoxic conditions produce and release high free MG levels, which we confirmed for non-transformed normal fibroblasts, albeit at significantly lower levels. Cancer cells generally significantly increased their free MG production and release when cultured in high glucose concentration, while normal cells generally did not. Furthermore, in normoxic conditions, normal fibroblastic cells, in addition to free MG, may produce and release lactate. From this data, we propose that in malignant tumors, both cancer and fibroblastic stromal cells may contribute to tumor growth and development by producing via glycolysis both free MG and D-lactate, which, in addition to L-lactate, may be part of the core hallmark of cell metabolic reprogramming in cancer.

## 1. Introduction

It is well known that malignant tumors are heterogeneous, particularly solid tumors, which encompass cancer cells and stromal cells, including cancer-associated fibroblasts (CAFs) and endothelial cells, mesenchymal cells and possibly inflammatory and immune cells such as macrophages and lymphocytes, among others [1,2,3]. Indeed, metabolism and proliferation of these different categories of tumor cells closely depend on the intra-tumoral microenvironment, which mainly relates to neoangiogenesis, a vascular process needed for the tumor to grow and develop [4,5,6,7]. An important observation is that neoangiogenesis vascularizes tumors via the peripheral stroma. This explains why glucose and oxygen levels within a solid tumor vary both spatially and temporally. Tumor cells located in the well-vascularized areas are better supplied with glucose and oxygen than cells in the poorly vascularized areas, and consequently, cells in the aerobic zones divide and proliferate, while cells in the hypoxic zones may exhibit an active metabolic quiescent state [8] before they may eventually undergo apoptosis and/or necrosis [9,10,11,12].

In recent years, cancer has emerged as a metabolic reprogramming process [13,14,15,16], with cancer cells being able to adapt and proliferate in different microenvironments. Although cancer cells prioritize aerobic glycolysis, a process referred to as the Warburg effect [17], to rapidly produce energy in the form of adenosine triphosphate (ATP) and to drive metabolic intermediates for biomass building [18,19], recent advances reveal that mitochondrial oxidative phosphorylation (OXPHOS) is also used by these cells to produce sufficient ATP for tumor growth and development [20,21]. Using a computational model, it has been recently proposed that normal cells could be associated with either one of these two following stable steady states: one which corresponds to the glycolytic phenotype, and the other to the mitochondrial OXPHOS phenotype. On the other hand, cancer cells may exhibit both processes, corresponding to a hybrid metabolic phenotype [22,23]. The upshot is that, unlike normal cells, cancer cells adapt to their microenvironment well enough to proliferate.

Indeed, due to genetic mutations and/or epigenetic modifications, a basic property of cancer cells is to overexpress glucose transporters and glycolytic enzymes [15,24,25], hence to upregulate glucose uptake and glycolysis [13,18,26]. Consequently, it appears that a side-product of glycolysis, methylglyoxal (MG), may be a metabolic biomarker whose production levels can discriminate between cancer cells and cells of normal tissues.

It is well known that during glycolysis, MG formation results from the spontaneous dephosphorylation of triose phosphate [27,28]. Due to its high electrophilic reactivity, MG glycates intracellular and extracellular macromolecules such as proteins [29] lipids and nucleic acids [30], forming advanced glycation end products (AGEs). Thus, to counteract and limit this harmful electrophilic effect of MG, mammalian cells have developed several cytosolic detoxifying enzymes, among which glyoxalase I and glyoxalase II constitute the major MG-inducible detoxification system [31].

Using a rat chemically-induced colon adenocarcinoma grafted model [32], we previously showed that following the subcutaneous administration of a tumorigenic cell clone, free MG blood levels were significantly higher in these animals than in animals grafted with a non-tumorigenic cell clone derived from the same tumor and that these higher free MG blood levels positively correlated with tumor growth; while in animals grafted with the non-growing tumor cell clone, free MG blood levels remain normal [33].

In humans, we also recently showed that free MG levels measured in the peripheral blood of cancer patients are significantly increased in comparison with normal subjects used as controls, and that these increased free MG levels correlate significantly with tumor growth and development. Hence, free MG measured in the peripheral blood of cancer patients constitutes a metabolic new clinically useful biomarker of tumor growth that allows prognosis assessment, therapeutic monitoring, and follow-up of patients [34].

Moreover, using free MG measurements in tumors of patients with non-small cell bronchus carcinoma we also showed that free MG can be recovered from the tumors at significantly higher levels than from their corresponding normal lung tissue; and that free MG levels from the tumors correlate positively with free MG levels in the peripheral blood, suggesting that free MG produced by tumoral cells is released into the peripheral blood of patients [34].

However, we know that free MG is also detected in normal animal cells [35] and that free MG is produced in peripheral blood of non-cancer diabetic patients [36,37] at significantly higher levels than in healthy subjects [37,38]; a finding that we confirmed in diabetic patients without cancer [34]. Consequently, the present research objective was to determine whether free MG in tumors is exclusively produced by cancer cells or whether it can also be produced by non-cancerous cells.

To answer this question, we measured free MG in different cancer and normal cell lines cultured in different microenvironmental conditions: low or high glucose concentration medium and normoxic or hypoxic conditions, to clarify whether these cells have the capacity to produce and release free MG. Since we have shown in this study that, like cancer cells, normal cells grown in normoxic conditions are able to produce and release free MG in the culture medium, we also measured lactate in the culture medium to verify whether normal cells could simultaneously produce and release lactate originating from free MG detoxification. Indeed, if we could show that lactate is produced by normal cells grown in normoxic conditions, this would reinforce our hypothesis that free MG may be produced and released in tumors by non-cancerous cells.

In this study, we find (1) that free MG is produced and released both from cancer and normal fibroblastic and endothelial human cells cultured in normoxic conditions; (2) that cancer cells, when cultured in high glucose concentration, generally significantly increase their free MG production and release in the culture medium, while normal cells cultured in the same conditions generally do not; (3) that simultaneously to the production and release of free MG, normal cells cultured in normoxic conditions produce and release low lactate levels; and (4) that the increased free MG and lactate production in low or high glucose concentrations culture medium mainly depends on the variety of cancer or normal cells investigated.

Finally, our current data leads us to hypothesize how free MG and lactate produced and released by cancer and non-cancer cells may play a distinct critical role in tumor growth and progression.

## 2. Material and Methods

### 2.1. Cell Cultures

We cultured a panel of human cancer cell lines: the MDA-MB-231, MDA-MA-468 and MCF-7 cell lines, originated from breast carcinomas; the U251 and U87 cell lines originated from glioblastomas, the H1299 and A549 cell lines originated from lung carcinomas, and the PC3, HCT116 and BX-PC3 cell lines originated from a prostate, colon and pancreas carcinoma, respectively. In addition, we used the HL60 cell line obtained from a human with promyelocytic leukemia.

We also used five normal human cell lines, one originated from dermal fibroblasts (NDF) and the other from normal osteoblasts (NHO); while the three other normal human cell lines, the MCF-10-A, SF5 and HUVEC, originated from epithelial breast cells, cutaneous fibroblasts and umbilical cord-derived endothelial cells, respectively. All cell lines were purchased from the American Type culture collection (ATCC, Manassas, VA, USA).

In this study, we used normal fibroblastic and endothelial cell lines because of their similar cell type with corresponding stromal cells, and we used normal epithelial cell lines to compare them to cancer epithelial cell lines for free MG production.

Because we had found that normal cell lines can produce and release free MG at higher levels than cancer cell lines, we also used primary cultures of normal fibroblasts to clarify whether non-transformed normal fibroblasts in culture could also produce and release free MG. We isolated and cultured normal fibroblasts from the breast of two patients who underwent a surgical mammary reduction for a non-cancerous breast hypertrophy (these two primary cultures were termed NBF-1 and NBF-2), and from the skin of a normal subject (this primary culture was provided by the Nutrition, Genetics and Environmental Risk Exposure (NGERE) of the Inserm-Lorraine-Nancy University laboratory (France) and termed HDF-DJO) to study MG intracellular production and release in the culture medium. We also used cultures of the PCS 201-012 normal dermal fibroblasts and the CCD-18CO normal colon fibroblasts purchased from ATCC.

Cancer and normal cells from either cell lines or primary cultures were cultured in plastic vials in Dulbecco’s Modified Eagle Medium supplemented by 10% embryonic calf serum and glucose at two different concentrations: one a low glucose concentration of 5 mM (1 g/L); and the other a high glucose concentration of 25 mM (5 g/L) because they are commonly reported in the scientific literature.

Before being tested for free MG and/or lactate release in cultures, all cell lines were adapted to grow in the low or high glucose concentration medium for 2 weeks. Then cells were grown in low or high glucose concentration medium for 48 h and counted. The derivatization step was done directly in the plastic vials before the resulting derivatized culture medium was collected for free MG and lactate measurements.

For each cell culture grown in low or high glucose concentration medium, three different types of experiments were performed in triplicate and free MG and lactate measurements were done each time in duplicate. Mean free MG levels were expressed in nmole per 10^6^ cells in culture and lactate in mmol per 10^6^ cells. All cell cultures were done in normoxic conditions, i.e., typically at 37 °C with 20% O_2_ and 5% CO_2_. In a second set of experiments, cells were cultured in hypoxic conditions (37 °C, 1% O_2_ and 5% CO_2_) and compared to normoxic conditions to investigate the influence of hypoxic conditions on free MG and lactate production by normal and cancer cells cultured in low glucose or high glucose concentration medium.

### 2.2. Use of Anti-MG Specific Probe

Because we wanted to know whether non-transformed normal fibroblasts could produce and accumulate MG intracellularly at a detectable level, we used a fluorescent-specific anti-MG probe kindly supplied by Dr. D. Spiegel (Yale University, New Haven, CT, USA). We studied the NFB-1 and NFB-2, and the PCS 201-012 and CCD-18CO non-transformed fibroblasts, as well as fibroblastic cells of transformed normal cell lines. The fluorescent sensor was methyl-diaminobenzene-BODIPY (or MBo), which can detect MG specifically under physiological conditions according to the “turn on” fluorescent sensor method described by Tina Wang et al. [39].

### 2.3. Free MG and Lactate Measurements

Free MG measurement in culture medium was done according to the protocol described by Rabbani and Thornalley in 2014 [40], and lactate measurement according to the method described by Tan et al. [41].

For free MG and lactate measurements, the derivatization procedure used 1,2-diaminobenzene or benzylhydroxylamine, respectively. Liquid chromatography was coupled to tandem mass spectrometry. Both methods are detailed in the Appendix A.

### 2.4. Statistical Tests

The two-tailed Student’s *t*-test was used for comparison between culture conditions, with the α significance cut-off at 0.05. When the two-tailed Student’s test was used to perform more than one comparison (c, for example), the Bonferroni correction was applied, which sets the α cut-off of significance at 0.05/c.

All statistical analyses were performed using the XLSTAT software Addinsoft (XLSTAT 2018.1.49725).

## 3. Results

Results are reported in the following figures, while raw data and their precise statistical significance are reported in the Appendix A.

### 3.1. Free MG Production and Release by Cancer Cell Lines Cultured in Normoxic Low or High Glucose Concentration

Figure 1 discloses our results for the different human cancer cell lines cultured in normoxic conditions and low or high glucose concentrations.

All cancer cell lines were found to be glycolytic since all produce and release free MG. Except for the two MCF-7 and BXPC3 cancer cell lines, which were derived from a breast and a pancreatic carcinoma, respectively, all investigated cancer cell lines (originated from different varieties of cancers: breast, colon, lung, brain and prostate) significantly increased their free MG production and release levels in the culture medium when grown in the high glucose enriched medium. This suggests that the majority of cancer cell lines have lost their normal glucose uptake and glycolytic regulation, with the exception of the two MCF-7 and BXPC3 cancer cell lines, which appear to have retained a certain capacity of normal regulation for glucose uptake and glycolytic activity.

Note that the two MDA-MB breast cancer-derived cell lines have lost their normal glucose regulation, while the MCF-7 cell line, which was also derived from a breast carcinoma, has not. As indicated above, this is also the case with the only pancreas cancer-derived cell line we have investigated (BXPC3).

### 3.2. Free MG Production and Release by Normal Cell Lines Cultured in Normoxic Low or High Glucose Concentration

Figure 2 discloses our results for the different human normal cell lines tested. Surprisingly, normal fibroblastic cell lines such as NDF and SF5 cultured in normoxic conditions exhibit high levels of free MG production and release in the culture medium, which exceed those obtained with most cancer cell lines reported in Figure 1. This was also the case for the HUVEC normal endothelial cell line. However, unlike most cancer cell lines, all normal cell lines tested, except the NHO normal cell line originated from normal osteoblasts, did not significantly increase their MG production and release in high glucose-concentration medium. This suggests that normal cell lines cultured in normoxic conditions may generally regulate their glucose uptake and glycolysis, while most cancer cell lines may not.

This result is in contrast with the finding of our previous bioclinical study, in which tumor cells in cancer patients produce and release significantly higher free MG levels than cells of their corresponding normal tissue [34]. In the present study, the fact that normal cell lines exhibit free MG high levels exceeding those obtained in cancer cell lines, may be due to a culture-transforming effect associated with normal cell lines (see further).

### 3.3. Intracellular MG Production and Free MG and Lactate Release by Non-Transformed Normal Fibroblasts

To understand whether the high production and release of free MG by normal cell lines could be due to the transforming effect of normal cells into permanent cell lines in culture, we looked for the intra-cellular production and accumulation of MG in primary cultures of non-transformed normal human fibroblasts, and measured free MG and lactate release in the culture medium of the PCS 201-012 non-transformed normal fibroblastic cells grown in normoxic or hypoxic conditions in low or high glucose concentration. The results are depicted in Figure 3, Figure 4 and Figure 5.

Figure 3 shows the fluorescent staining we obtained with the use of the specific anti-MG probe on NBF-1 and NBF-2 normal fibroblasts of primary cultures originated from the two non-cancer breast tissue samples and on cells of the NDF transformed normal cell line originated from dermal fibroblasts, all cultures having been done in normoxic low or high glucose concentration. Similar results were obtained with the PCS 201-012 and the CCD-18CO normal fibroblast cultures. This reveals that intracellular MG can be detected even in non-transformed normal fibroblasts.

Furthermore, we measured the level of free MG and lactate production and release by PCS 201-012 non-transformed normal fibroblasts cultured in normoxic or hypoxic conditions, and low or high glucose concentration. Figure 4 summarizes our data. Free MG and lactate are released in the culture medium from PCS 201-012 normal fibroblasts in either normal aerobic or anaerobic conditions in low or high glucose concentration.

The finding that the lactate produced and released by normal cells occurs in normoxic conditions, not just in hypoxic conditions, suggests that normal fibroblasts do not restrict their lactate production to the conversion of pyruvate into L-lactate when they are cultured in hypoxic conditions but produce D-lactate from free MG detoxification by the glyoxalase main detoxification enzymatic system, even in normoxic conditions.

However, as indicated in Figure 5, levels of free MG released from the PCS 201-012 and the HDF-DJO non-transformed normal fibroblasts cultured in normoxic conditions in low or high glucose concentration are significantly much lower than those released from the corresponding NDF and SF5 transformed normal fibroblastic cell lines. The comparison between the results obtained from the two different types of non-transformed normal fibroblasts and the two transformed normal fibroblastic cell lines investigated for free MG production and release in the culture medium in normoxic conditions suggests that the significantly higher free MG produced and released by normal fibroblastic cell lines may have been due to the culture transformation process indicated above.

### 3.4. Free MG Production and Release by Normal Fibroblasts and Cancer Cell Lines Cultured in Normoxic or Hypoxic Conditions in Low or High Glucose Concentration

Figure 6 exemplifies our data, showing that free MG production and release by the U251 glioblastoma cell line is significantly lower than that by the PCS 201-012 non-transformed normal fibroblasts, whatever the oxygen and glucose conditions. But it clearly appears that the free MG production and release by the U251 cell line and the PCS 201-012 normal fibroblasts strongly contrasts with the significantly higher free MG production and release by the HL60 leukemic cell line cultured in either normoxic or hypoxic conditions in low or high glucose concentration, and it is again significantly much higher for the U87 cell line cultured in normoxic or hypoxic conditions, but only for the high glucose concentration. As exemplified with the U87 glioblastoma-derived cell line, an important point outlined once again is the extreme dependence on glucose concentration of free MG production and release by cancer cells.

### 3.5. Lactate Production and Release by Normal and Cancer Cell Lines Cultured in Normoxic Low or High Glucose Concentration

As represented in Figure 7A,B, the two SF5 and HUVEC normal human cell lines (derived from fibroblastic and endothelial cells, respectively) increased their lactate production very weakly and release it in low or high glucose concentration, while the normal MCF-10A epithelial breast cell line almost did not. Note that in comparison with the three normal cell lines investigated, the U87 and HL-60 cancer cell lines strongly increased their lactate production and release in the culture medium, while this production and release by the U251 cell line—although it was also derived from a human glioblastoma—did not significantly differ from the lactate production and release by the SF5 normal cell line. Note also that the U87 and U251 cancer cell lines increased significantly their lactate production when cultured in a high glucose concentration medium, whereas the HL-60 leukemic cell line did not.

Altogether, our results confirm that in comparison with normal cell lines, cancer cell lines cultured in aerobic conditions generally increase significantly their lactate production—a finding which therefore confirms the Warburg effect. However, our results also suggest that in addition to their much higher lactate production and release in the culture medium, cancer cells may exhibit different metabolic behaviors in culture. For example, compared to the SF5 fibroblastic normal cell line, the U251 glioma cell line produced and released similar lactate levels. Therefore, our findings confirm that the Warburg effect identified by lactate measurements in aerobic conditions (aerobic glycolysis) applies globally to cancer cells, but their lactate production and release may vary in amplitude, according to the variety of cancer cells and the glucose concentration. Importantly, the Warburg effect measured at cellular levels may also apply to normal cells, but generally at much lower levels, a finding confirming that the Warburg effect is not specific to cancer cells.

### 3.6. Lactate Production and Release by Cancer Cell Lines Cultured in Hypoxic or Normoxic Conditions and Low or High Glucose Concentration

Figure 8A,B summarize our data. It exemplifies that in hypoxic conditions compared to normoxic conditions, the U87 cell line significantly increases its lactate production and release when cells are cultured in low glucose concentration (Figure 8A) but not when cells are cultured in high glucose concentration (Figure 8B); while the U251 cell line increases significantly its lactate production and release when cells are cultured either in low or high glucose concentration (Figure 8A compared to Figure 8B). In contrast, the HL60 leukemic cell line does not significantly increase its lactate production and release in either low or high glucose concentration. Here too, it appears that among the three different cancer cell lines investigated so far, there is some variability in their in vitro behavior, depending on oxygen and glucose culture conditions.

## 4. Discussion

The idea that tumor growth depends not only on cancer cells but also on the tumor microenvironment (TME)—mainly on stromal CAFs, the most abundant TME cellular component—is a new emerging scientific concept [14,42,43]. Indeed, it is increasingly recognized that TME plays a major role in tumor invasiveness, progression, metastasis, and epithelial–mesenchymal transition [44,45,46]. This is the case with stromal CAFs, which have been shown to interact with cancer cells, to initiate extracellular matrix remodeling [47] and to achieve tumor progression with [48] or without [49] increased neoangiogenesis.

Along with the presumed symbiotic tumoral interplay between cancer cells and stromal cells, the metabolic reprogramming in tumor cells greatly contributes to tumor growth. Clearly, abnormal metabolism is an emerging hallmark of cancer [13,15,50], and glucose is the key molecule to be taken up and metabolized by cancer cells at a higher rate than normal cells to rapidly generate ATP and to synthetize new biomass via glycolysis-derived metabolites [20,51].

Because glucose uptake and glycolysis occur faster than OXPHOS [52], cancer cells can increase anaerobic glycolysis in the presence of oxygen availability. The process where tumor tissue achieves a higher increase in glucose consumption and lactate production in aerobic conditions than normal tissue was finally conceptualized and published in 1956 by Otto Warburg and termed aerobic glycolysis [17].

We now know that aerobic glycolysis is associated with an aggressive metabolic phenotype because cancer cells produce ATP at a faster rate and because they contribute greatly to biomass building that is required for tumor cell proliferation and tumor growth [19]. However, although cancer cells may utilize primarily aerobic glycolysis, recent evidence suggests, as reported above [22,23], that they also use OXPHOS and the mitochondrial electron-transport chain (ETC) respiration to adapt to their microenvironment and to produce tumor invasion, progression and metastasis [21,53,54].

In fact, there is a metabolic heterogeneity in tumors, with some cells remaining at a glycolytic state, and some others using OXPHOS predominantly [53,55]. This is the case for cancer cells as well as for stromal cells. This explains why oxidative cancer cells generate great quantities of ATP to proliferate, while glycolytic cancer cells may still undergo a predominant Warburg effect by producing and releasing high quantities of lactate, which may be incorporated and metabolized by all proliferating tumor cells for their energetic needs. Indeed, during this process, cancer cells that release mitochondrial reactive oxygen species (ROS) may contribute to activating stromal CAFs by promoting their metabolic reprogramming, hence to also producing lactate to supply their own energetic needs and to fuel cancer cells in the framework of the so-called reverse Warburg effect [56,57].

In this study we showed that cancer cells cultured in normoxic conditions generally produce and release free MG in addition to significantly higher levels of lactate than do normal cells cultured in the same conditions, a finding which confirms the Warburg effect at cellular levels; furthermore, we showed that in normoxy, normal cells are also able to produce and release free MG and low lactate levels in the culture medium.

Since we previously showed that free MG can be recovered from human tumors and be released in the peripheral blood of cancer patients at levels that correlate significantly with tumor growth [34], we propose that in tumors, both cancer cells and stromal cells may produce free MG and lactate, and release into the extracellular TME these two intra-tumoral molecules, which we and others proposed to be major metabolic effectors, contributing differentially to tumor growth and development.

### 4.1. Free MG Production by Cancer Cells and Non-Cancer Cells

In the present study, we found that not only cancer cells but also normal cells cultured in normoxic conditions produce and release free MG in the culture medium. This has been observed for cells of the normal fibroblastic and endothelial cell lines, and for non-transformed normal fibroblasts in culture. The very high free-MG levels released by normal cell lines contrast with the results we previously obtained from our cancer patients’ bioclinical study in which free MG levels in the peripheral blood of normal subjects do not exceed 0.06 µM, while free MG levels in cancer patients exceeded this reference value significantly and were significantly correlated with tumor growth [34]. As discussed above, the high free MG production and release by normal cell lines may have been due to a transforming effect of normal cells into permanent cell lines in culture. Moreover, we showed that non-transformed normal fibroblasts can also produce free MG, but at significantly lower levels than normal cell lines. We therefore interpret the free MG normal reference value found in healthy controls in our previously reported cancer patients study as resulting from the fact that in vitro non-transformed normal fibroblasts produce and release significantly lower free MG levels than cancer cells; and from that in vivo, tightly regulated cells within normal tissues may limit their glucose uptake and glycolysis-related free MG production, whereas cancer cells do not.

Such higher free MG production by cancer cells or by transformed normal cells in culture may be due to the inhibiting effects of endogenous and/or exogenous causal agents on the activity of glyceraldehyde 3-phosphate dehydrogenase (GAPDH), which normally oxidize glyceraldehyde-3-phosphate to 1,3-bisphosphoglycerate. Since deficient activity of this enzyme may result in increased free MG production by both cancer or transformed normal cells in culture, it appears that excess free MG results from a pathological process of glycolysis. This may be due to an initial inhibiting activity of this enzyme by ROS, then by its covalent combination with reactive free MG as a subsequent inhibiting event, causing amplification of this loss of activity due to the formation of AGEs [58]. Note that this might not only be the case in cancer patients but also in diabetic patients, in whom ROS [36] and MG [37] have been shown to impair the insulin pathway.

Again, it is well known that due to its electrophilic properties, MG is a highly reactive dicarbonyl glycating agent of intracellular and extracellular macromolecules, leading to AGEs such as MG-hydroimidazolones and MG-Argypirimidines, which have been shown to be commonly formed in cancers [30,59]. An important consequence of the increased glycolytic activity in tumor cells (whatever the oxygen concentration in tumors) is that the major part of the intracellular MG reacts with macromolecules to form AGEs, while a lower part of it is released from tumors in the form of measurable free MG in the host extracellular compartment [60]—possibly when the intracellular-increased AGEs formation has reached a maximum threshold value because of the rate-limiting enzymatic detoxification processes. Indeed, since intracellular AGEs have been shown to exhibit a dual role in cancer progression [61], we have focused the present study on free MG, which we have shown here to be produced and released by cancer and non-cancer cells, and which we had previously shown to be a marker of tumor growth [33,34].

### 4.2. Free MG and Lactate as Permeable and Diffusible Molecules

Since they are released in the culture medium, we confirmed in this culture study that both glycolysis-derived free MG and lactate are permeable and diffusible molecules. This clearly requires consideration of the different transmembrane cell transporters. For lactate and other monocarboxylate molecules, the different types of monocarboxylate transporters (MCTs) have been considered critically necessary for their cell release and uptake.

Unlike the relatively well-defined MCT-associated transportation of lactate, the transmembrane export of free MG from cancerous or non-cancerous cells remains problematic since, to our knowledge, there is no reported data showing that it may involve an export dicarboxylate transporter [62]. A hypothetical possibility is that it could involve a metabolic cargo using extracellular vesicles [63,64], but this has yet to be shown. Clearly, free MG transmembrane export must differ from the MCTs involved in monocarboxylate molecules such as lactate.

Regarding free MG release, our study suggests that distinguishing cancer from normal cells may be possible, since cells from most cancer cell lines cultured in normoxic conditions significantly increase their free MG release in the culture medium when grown in high glucose concentration, while cells from most normal cell lines do not. As discussed above, this may be due to the fact that normal transformed cell lines have retained a certain degree of glucose regulation, whereas cancer cell lines have not. However, this suggestion deserves further scientific debate since as depicted in Figure 4 we found that non-transformed normal fibroblasts may also significantly increase their free MG production and release when grown in a high glucose concentration medium in either normoxic or hypoxic conditions. This suggests that in vitro, normal cells may also lose part of their glucose regulation.

### 4.3. On the Non-Specificity of the Warburg Effect Applied to Isolated Cultured Cells

Regarding lactate release, this study confirms that cells from cancer cell lines cultured in normoxic conditions in low or high glucose concentration increase their lactate production and release in the culture medium generally at significantly higher levels than normal cells do. This was the case for two out of the three cancer cell lines, which produced and released significantly higher lactate levels than normal cell lines. Therefore, this finding confirms that the Warburg effect, which was shown by Warburg to apply to tumor tissues, also applies to isolated cancer cells grown in culture, but it does not apply specifically to cancer cells. It is remarkable that in this study, one normal cell line investigated increased its lactate production and release in aerobic conditions at similar levels as did one of the three cancer cell lines so far investigated. This suggests that, as for cancer cells, normal cells grown in aerobic conditions may also produce some D-lactate coming from the intracellular MG detoxification (see further).

We know that Warburg hypothesized that the effect he described was specific to tumor tissues [17] and possibly caused by an enzymatic defect in the mitochondrial OXPHOS and/or ETC respiration in cancer cells [65]. Although the Warburg effect has been documented in many tumors, including breast, bronchus and colorectal carcinomas and melanomas [66], such a mitochondrial enzymatic defect was not found to specifically cause an increased aerobic glycolysis in most investigated cancer cases [53]. Rather, it was clearly established that the mitochondrial enzymatic dysfunction observed in cancer cells may allow these cells to release ROS into the TME [67]. This generates oxidative stress in neighboring CAFs, which were shown to contribute to tumor growth and metastasis [67,68,69,70]. In fact, under the influence of hypoxia, both cancer cells and activated CAFs produce ROS such as hydrogen peroxide (H_2_O_2_) [71]. Furthermore, ROS produced by cancer cells were thought to play a pivotal role in CAF activation [72]. Accordingly, oxidative stress in CAFs was considered to be a core part of the reverse Warburg effect. Such non-specificity of the Warburg effect had already been observed in different normal proliferating human cells, for example, in embryonic stem cells [73,74] and in activated T lymphocytes [75,76].

### 4.4. High Methylglyoxal Production as a New Core Hallmark of Cell Metabolic Reprogramming in Cancer

Although evidenced to be characteristic but not specific to cancer cells, the Warburg effect questions the type of lactate produced and released by cancer cells during aerobic glycolysis. There are indeed two glycolysis-related main mechanisms for lactate production: one that relies on free MG, which is normally mainly detoxified by the glyoxalase system and leads to D-lactate [77]; and another that results from the conversion of pyruvate into L-lactate by type A lactate dehydrogenase (LDH-A) [25]. In this study, we did not measure the two lactate isomers, so we cannot objectively answer the preceding question.

However, we showed in the present study that cancer cells cultured in normoxy are associated with the production and release of free MG, which we [34] and others [69,78,79] interpreted as resulting from a saturation and/or a weakening of the rate-limiting glyoxalase and other free MG detoxification inducible enzymes, allowing cancer cells to reach a maximum intracellular glycation threshold value, thereby releasing free MG from cells. Since the glyoxalase detoxification enzymatic system may have contributed to detoxifying part of the intracellular free MG accumulation, we believe that during aerobic glycolysis part of lactate produced by cancer cells and by other tumor cells may not only result from the conversion of pyruvate into L-lactate as it is commonly reported but also from the production of D-lactate produced by the glyoxalase free MG detoxification process. Increased amount of D-lactate has been evidenced in breast [80] and prostate carcinoma cells [81] as well as in lung cancers [82] to such a degree that free MG and D-lactate levels, in addition to L-lactate, should now be considered a core hallmark of tumor cells’ metabolic reprogramming [83].

These findings open a new way of thinking for research. Since, according to the concept of a reverse Warburg effect [56], lactate released by activated CAFs has been proposed to fuel OXPHOS in aerobic cancer cells, and consequently to contribute to tumor growth, it remains to be determined what quantity of D-lactate relative to L-lactate could be produced by cancer cells.

Indeed, the large quantity of lactate produced by glycolytic cancer cells and released into the TME could fuel not only cancer cells via an autocrine loop but also oxidative stromal CAFs via a paracrine way [84]. Likewise, lactate produced by activated CAFs may fuel oxidative cancer cells and oxidative stromal cells in a paracrine way. This fuel effect can occur through the conversion of L-lactate into pyruvate by the lactate dehydrogenase B (B-LDH). For D-Lactate, it could occur through the conversion of D-lactate to pyruvate by D-LDH, a stereochemical LDH enzyme, which has been isolated and purified [85] and found in human tissues with high metabolic rate, and in cancer [86]. It is possible that, as reported in *lactobacillus plantarum* [87], L-lactate, pyruvate and D-lactate might be in cells in a reversible equilibrium as follows:L-lactate ⇋ pyruvate ⇋ D-lactate

Be that as it may, it appears that in addition to the proposed reverse Warburg effect involving activated stromal CAFs, both D and L lactate produced by cancer cells in tumors may significantly contribute to fueling oxidative cancer and stromal cells for tumor growth.

### 4.5. Free MG and Lactate as Intra-Tumoral Molecular Effectors That May Contribute Distinctively to Tumor Growth and Development

Our previous findings that free MG is associated with tumor growth and can be recovered from tumors [34], in addition to Warburg’s seminal observation that lactate is recovered from tumors [17] and that it may also be associated with tumor growth (see further), lead us to consider that in tumors cancer cells and stromal cells may produce and release not only lactate, but also free MG, and that both intra-tumoral molecular components may act differentially to drive cancer cells to proliferate and tumor to progress. This indeed may occur if tumors are sufficiently supplied with glucose.

Since, in the present study, we show that in normoxy, all cancer cell lines, as well as normal fibroblastic and endothelium cell lines and non-transformed normal fibroblasts, produce and release free MG, and that in such conditions cancer cell lines produce and release high quantity of lactate, while normal cell lines and normal fibroblasts generally produce and release significantly lower lactate levels, we hypothesize that in tumors, both cancer cells and stromal cells may produce and release free MG into the TME, while cancer cells may not produce and release the total part of lactate but the largest part of it.

Since we have shown in this study that normal fibroblasts may produce and release free MG and lactate, our further hypothesis is that host-derived normal stromal fibroblasts in tumors may coexist with cancer cells and activated glycolytic CAFs to produce both intratumoral molecular components.

Clearly, normal stromal fibroblasts should be distinguished from cancer cells-activated stromal glycolytic CAFs, which have been reported to produce and release lactate to fuel oxidative cancer cells and to promote tumor growth and development [48,88,89]. Indeed, under the oxidative stress attack induced by cancer cells, the reverse Warburg effect has been proposed to result from a loss of caveolin-1 in stromal fibroblasts, or to an OXPHOS downregulation in these cells with consequent glycolytic predominance [90].

As referred to in an initial definition [91] applied to malignant tumors, the transportation of lactate from cancer cells to neighboring stromal cells and vice versa has been termed “lactate shuttles”. In this context of metabolic cell reprogramming [92] lactate shuttles were thought to allow cancer cells to fuel neighboring stromal cells by coupling their Warburg’s glycolytic activity with these stromal cells [18]; whereas more recently, via the reverse Warburg effect, it was proposed that activated stromal CAFs may fuel cancer cells by coupling their own glycolytic activities with those of oxidative cancer cells [93].

We believe that this concept, mainly issued from experimental co-cultures, may not apply to the true in vivo situation in tumors, since according to our culture study, lactate may be produced mainly by cancer cells but also to a lesser degree by normal fibroblasts, and consequently, that the lactate produced by these two types of cells and by the cancer cells-activated CAFs may be released and spread out across the entire TME extracellular milieu. Thus, any oxidative tumor cells may take up and metabolize lactate in the TME milieu, whatever their cellular origin.

In fact, the extracellular TME should be considered as a biological milieu in which ROS, free MG and D- and L-lactate may contribute to tumor progression. This is indeed not only the case for free MG and lactate (see further) but also for cytokines produced by tumor cells. Clearly, the hypoxic TME produces and promotes oxidative stress, because hypoxia is associated with mitochondrial ROS production by cancer cells and glycolytic acquisition by stromal cells [67].

From previously reported data, we propose that because of its acidity, the extracellular lactate (in addition to its presumed capacity to fuel cancer cells) may be cytotoxic, and induce apoptosis of host normal cells around tumors, and of normal stromal cells within tumors [94], thereby providing room for tumor cells to proliferate, for neo-angiogenesis to occur via upregulation of the vascular endothelial growth factors [95], and for metastasis to develop via enhancement of cancer cell mobility [96,97,98,99].

In addition, we know that in tumor hypoxic zones, the transcription of hypoxia-inducible factor-1α (HIF-1α) may activate the transcription of numerous genes involved in glucose uptake, glycolysis, and oxygen consumption, and may promote cancer cell invasion and migration [100]. Indeed, via HIF-1α activation, hypoxia also stimulates the expression in cancer cells of anti-apoptotic, proliferative and neo-angiogenic genes that clearly contribute to enhancing tumor growth and aggressiveness [101,102].

Differing from the lactate tumor growth-promoting effects, free MG, once it is glycated with the extracellular intra-tumoral molecular components, may activate RAGE, the multi-ligand receptor for AGEs, which has been shown to be expressed in both cancer and normal cells [103,104,105,106], and it has been shown that once it is activated, it can inhibit apoptosis [107] and promote tumor growth and metastasis via molecular pathways such as AP-I, NF-kB, PI3K and mTor [60]. RAGE activation is an important tumor growth and metastasis-promoting process since its inhibition has been shown to decrease tumor growth and metastasis [108].

### 4.6. Study Limitations

There are several limitations to the present study and its resulting hypothesis:

First, we cultured cells from cancer and normal cells that were not directly isolated from tumors, and we hypothesized that our in vitro findings may correspond to the in vivo behavior of cells in tumors. Indeed, concerning glucose and the Warburg effect, there could be some discrepancies between in vitro and in vivo biology [109]. As discussed above, transformed cell lines in culture may be characterized by transforming behavioral properties that differ from non-transformed cells and from cells in tumors. Furthermore, the normal fibroblasts that we investigated in this study clearly differ from the putative activated CAFs in tumors.

Second, we did not use stromal fibroblasts issued from tumors co-cultured with cancer cells to study free MG and lactate production and release, but rather cancer and normal cell lines and normal fibroblasts cultured alone. Consequently, this procedure did not provide us with the possibility to use transcriptomics to study the molecular interplays between cells.

Third, the behavior of cells in culture depends not only on the environmental culture conditions but also on the type of cells.

Finally, we addressed our hypothesis to glycolytic tumors, i.e., to tumors whose glycolytic cells produce and release high amounts of free MG and lactate in normoxy. Non-glycolytic tumors whose cells produce spontaneously low levels or almost no free MG and lactate, i.e., in which the Warburg effect is weak or even does not apply, may be associated with a different pathophysiological profile and with a different clinical outcome, and these may be associated with a better prognosis.

## 5. Conclusions

By using cultures of cancer and non-cancer cells, we further developed our observation of the significant increase in free MG levels in the blood of cancer patients. We reinforced our previous hypothesis that, in addition to cancer cells, stromal cells (particularly fibroblasts) may produce and release free MG and lactate in tumors. We now propose that not only lactate but also free MG produced and released from both cancer and stromal cells in tumors may distinctly contribute to tumor growth and progression. Further investigations are needed to validate this hypothesis.

## Figures and Tables

**Figure 1 cells-14-00931-f001:**
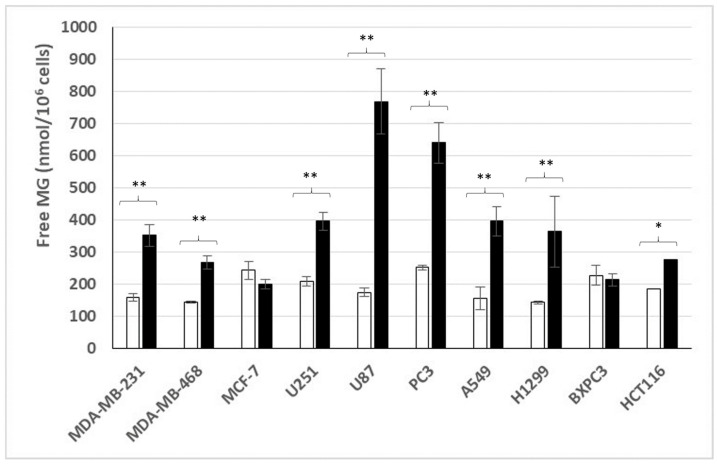
Free MG release from cancer cell lines cultured in normoxic low ☐ or high ⬛ glucose concentration. *: *p* = 0.001; **: *p* < 0.0001.

**Figure 2 cells-14-00931-f002:**
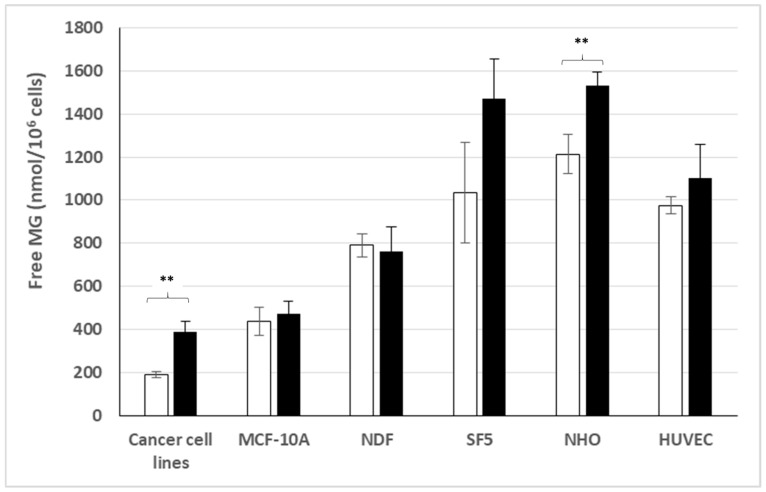
Free MG release from normal cell lines in comparison with mean free MG release from cancer cell lines, in normoxic low ☐ or high ⬛ glucose concentration. **: *p* < 0.0001.

**Figure 3 cells-14-00931-f003:**
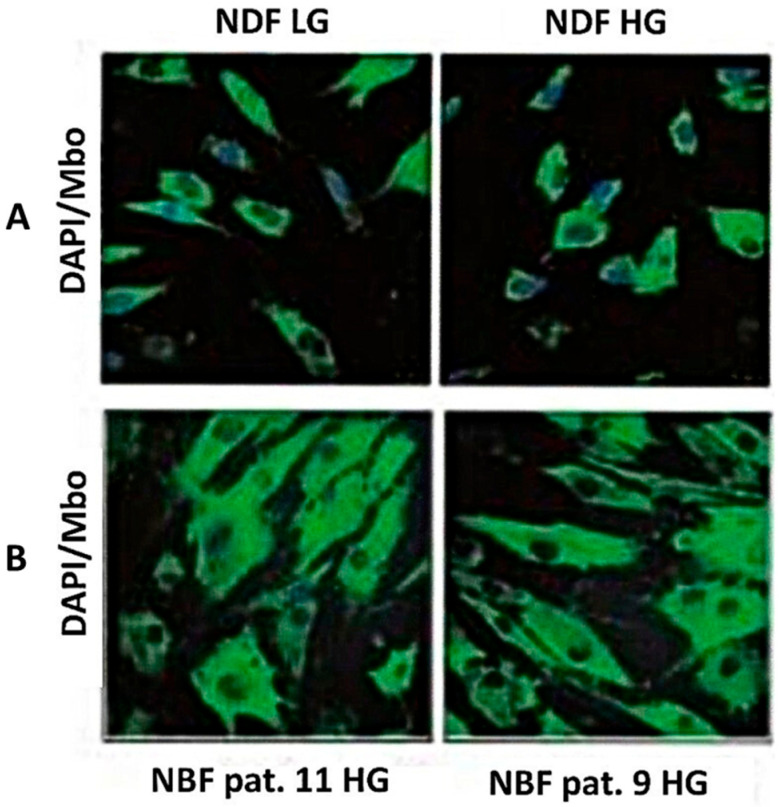
Fluorescent staining using a specific anti-MG probe (**A**) in NDF normal fibroblast line, and (**B**) NFB-1 and NFB-2 normal fibroblasts. LG: low glucose concentration; HG: high glucose concentration.

**Figure 4 cells-14-00931-f004:**
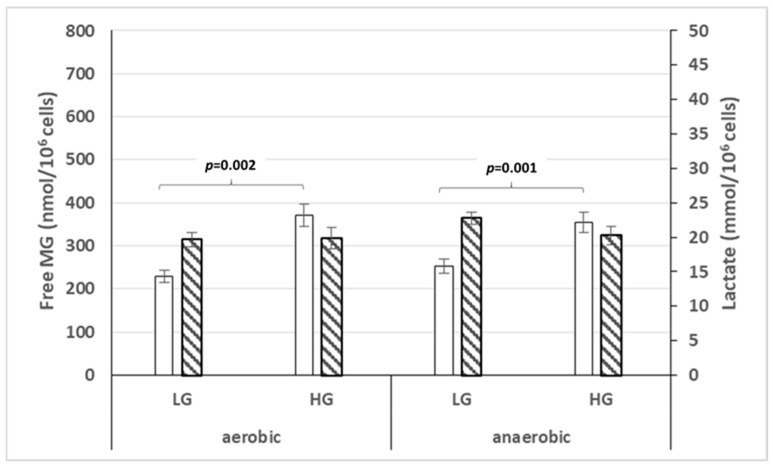
Representation of free MG ☐ and lactate ▧ release by PCS 201-012 normal fibroblasts in normoxic or hypoxic conditions and glucose concentrations. LG, low glucose concentration; HG, high glucose concentration.

**Figure 5 cells-14-00931-f005:**
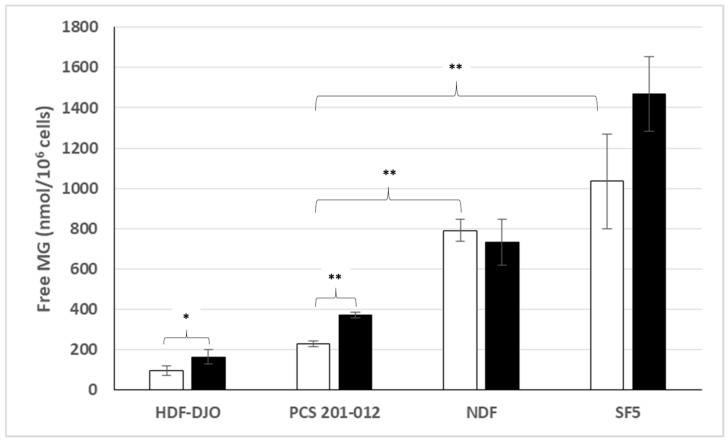
Representation of free MG release from non-transformed normal fibroblasts and normal fibroblastic cell lines in normoxic low ☐ or high ⬛ glucose concentration. *: *p* < 0.001; **: *p* < 0.0001.

**Figure 6 cells-14-00931-f006:**
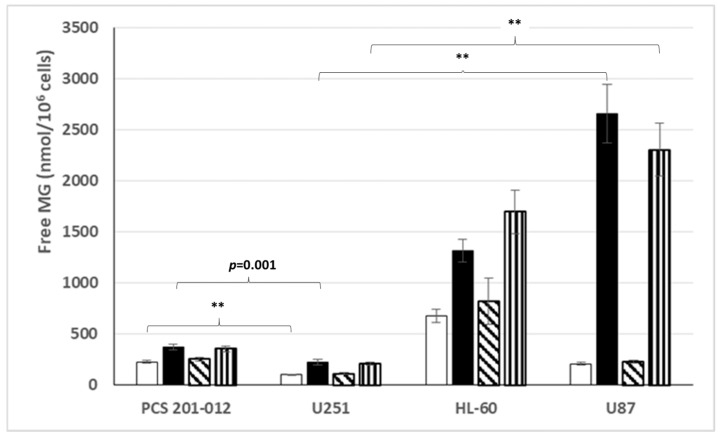
Free MG release from normal fibroblasts and cancer cell lines cultured in different conditions. Normoxic low glucose ☐, normoxic high glucose ⬛, hypoxic low glucose ▧, or hypoxic high glucose ▥ conditions. **: *p* < 0.0001.

**Figure 7 cells-14-00931-f007:**
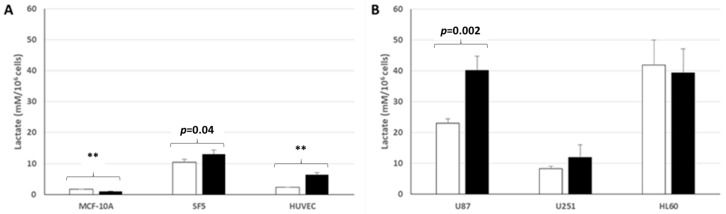
Lactate production from normal (**A**) and cancer cell (**B**) lines in normoxic low ☐ or high ⬛ glucose concentration. **: *p* < 0.0001.

**Figure 8 cells-14-00931-f008:**
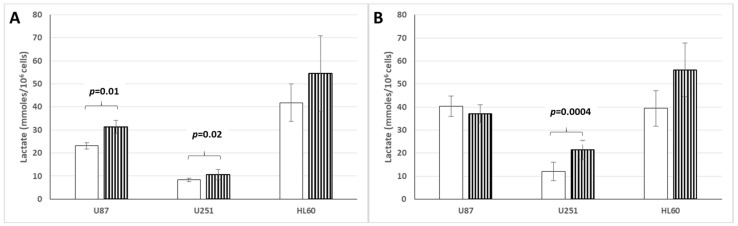
Lactate release from cancer cell lines in aerobic ☐ or anaerobic ▥ low (**A**) or high (**B**) glucose concentration.

## Data Availability

Study data can be made available to interested researchers upon request. Requests can be directed to data manager Philippe Irigaray (philippei.artac@gmail.com).

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
