# Peer review of "Free Methylglyoxal and Lactate Produced and Released by Cultured Cancer and Non-Cancer Cells: Implications for Tumor Growth and Development"

_cells, 2025, doi:10.3390/cells14120931_

Round 1
Reviewer 1 Report
Comments and Suggestions for Authors
This manuscript explores the differential effects of methylglyoxal (MG) and lactate on tumor and non-tumor cells under varying glucose conditions. While the study provides valuable insights into metabolic alterations in the tumor microenvironment, several aspects require improvement.
- Is the rationale for selecting these tumor and non-tumor cell lines clearly stated, particularly whether they possess known metabolic characteristics?
- Are the high and low glucose concentrations clearly defined, and is the rationale for choosing these concentrations explained?
- The observed variability in certain datasets (e.g., MG release from U87 cells in low glucose) necessitates additional biological replicates to ensure robust conclusions.
- The study does not differentiate between D-lactate and L-lactate isomers, potentially overlooking the significant contribution of D-lactate production via the glyoxalase-mediated detoxification pathway of methylglyoxal (PMID: 18946510).
- The cell models used, including cell lines and primary cells, do not fully replicate the complex interactions within the tumor microenvironment, particularly due to the absence of co-culture experiments involving cancer cells and stromal cells.
- Although elevated blood MG levels are associated with tumor progression (PMID: 39682111), the cellular sources (epithelial cancer cells vs. CAFs vs. endothelial cells) of MG in tumor tissues remain unverified. This gap limits the interpretation of MG's role in tumor-stroma crosstalk, particularly given that stromal cells may contribute significantly to the MG pool (PMID: 32190311). Future studies should employ spatial transcriptomics or cell-specific MG flux assays to resolve this.
- The study does not fully elucidate the mechanism of methylglyoxal (MG) release from cells, including whether it traverses the plasma membrane via passive diffusion, specific transporters, or exosome-mediated secretion. Emerging work on extracellular vesicle-mediated transfer of reactive metabolites (PMID: 34957539, 36469967, 39627188) may offer relevant conceptual frameworks for understanding MG dynamics in tumor microenvironments. We recommend discussing these potential mechanisms in the context of your findings.
- The statistical methods (e.g., t-test/ANOVA) should be explicitly stated in Materials and Methods, with significance levels (e.g., p<0.05, p<0.01) clearly labeled in figures.
- The number of independent replicates (n-value) for each experimental group should be explicitly stated, with data presented as mean ± standard error (SE) or standard deviation (SD) in all figures.
- In your manuscript, the 7. Conclusion section may have formatting errors, appearing incorrectly as “7. . Conclusion.”.
Author Response
Reviewer 1
This manuscript explores the differential effects of methylglyoxal (MG) and lactate on tumor and non-tumor cells under varying glucose conditions. While the study provides valuable insights into metabolic alterations in the tumor microenvironment, several aspects require improvement.
Response
In introduction, we would like first to stress the semantic difference between tumor cells and cancer cells, a distinction that you seem to have not considered: cancer cells should be clearly distinguished from tumor cells which encompass cancer cells and non-cancer cells, i.e. stromal cells.
Comment 1
Is the rationale for selecting these tumor and non-tumor cell lines clearly stated, particularly whether they possess known metabolic characteristics?
Response 1
We have precised the rationale for selecting non-cancer cell lines, by adding a specific paragraph. There was no specific rationale for selecting cancer cell lines. See lines 142 to 144.
Comment 2
Are the high and low glucose concentrations clearly defined, and is the rationale for choosing these concentrations explained?
Response 2
High and low glucose concentration were clearly defined in lines 156-160, and the rationale for choosing these two concentrations is that they are those referred in all papers published in the scientific literature. This is indicated in lines 159-160.
Comment 3
The observed variability in certain datasets (e.g., MG release from U87 cells in low glucose) necessitates additional biological replicates to ensure robust conclusions.
Response 3
Thank you for pointed this point. As indicated in material and methods, all three sets of culture experiments were done in triplicate and free MG and lactate measurements were done twice. This is indicated in the paper in lines 166 and 168. Consequently, the observed variation in certain data sets ensure robust conclusions.
Comment 4
The study does not differentiate between D-lactate and L-lactate isomers, potentially overlooking the significant contribution of D-lactate production via the glyoxalase-mediated detoxification pathway of methylglyoxal (PMID: 18946510).
Response 4
We agree that we did not distinguish between L- and D-lactate isomers. This point is discussed in the discussion section 4. This distinction is presently under investigation. It does note negate the general scientific conclusion of the paper.
Comment 5
The cell models used, including cell lines and primary cells, do not fully replicate the complex interactions within the tumor microenvironment, particularly due to the absence of co-culture experiments involving cancer cells and stromal cells.
Response 5
The objective of the paper was not to investigate the complex cell interactions within the TME, since several papers, using co-cultures have already investigated this. Published results of cocultures are discussed in the discussion section 5. The fact that we did not perform co-cultures is mentioned and discussed in study limits in the discussion section 6.
Comment 6
Although elevated blood MG levels are associated with tumor progression (PMID: 39682111), the cellular sources (epithelial cancer cells vs. CAFs vs. endothelial cells) of MG in tumor tissues remain unverified. This gap limits the interpretation of MG's role in tumor-stroma crosstalk, particularly given that stromal cells may contribute significantly to the MG pool (PMID: 32190311). Future studies should employ spatial transcriptomics or cell-specific MG flux assays to resolve this.
Response 6
We agree with this comment. Our study should be considered as a first approach. We agree that future study should use spatial transcriptomics and cell-specific MG flux assays. This is indicated in the discussion section 6.
Comment 7
The study does not fully elucidate the mechanism of methylglyoxal (MG) release from cells, including whether it traverses the plasma membrane via passive diffusion, specific transporters, or exosome-mediated secretion. Emerging work on extracellular vesicle-mediated transfer of reactive metabolites (PMID: 34957539, 36469967, 39627188) may offer relevant conceptual frameworks for understanding MG dynamics in tumor microenvironments. We recommend discussing these potential mechanisms in the context of your findings.
Response 7
We also agree that our study does not elucidate the mechanism of free MG release from cells. Due to its extreme reactivity with macromolecules, we do not believe that it can go through the plasma membrane via passive diffusion. So we thought it may act via still unknown specific transporters. We added a specific paragraph in the discussion section 2 on the hypothetic possibility of extracellular vesicle-mediated cargo transfer of reactive metabolites such as MG with the two references reviewer 1 provided us.
Comment 8
The statistical methods (e.g., t-test/ANOVA) should be explicitly stated in Materials and Methods, with significance levels (e.g., p<0.05, p<0.01) clearly labeled in figures.
Response 8
We have precised the statistical method used in the core material and methods section of the paper, lines 193-196.
Comment 9
The number of independent replicates (n-value) for each experimental group should be explicitly stated, with data presented as mean ± standard error (SE) or standard deviation (SD) in all figures.
Response 9
The number of independent replicates is indicated in materials and methods.
Comment 10
In your manuscript, the 7. Conclusion section may have formatting errors, appearing incorrectly as “7. Conclusion.”
Response 10
We have individualized the paragraph “Conclusion” by suppressing “7”.
Reviewer 2 Report
Comments and Suggestions for Authors
- The manuscript could benefit from a more careful editing for English language.
- Abstract, 1st paragraph: how do you get tumors in cancer-free patients? How do you show correlation between circulating levels of MG and tumor growth in cancer-free patients?
- Line 47: “these different categories of tumor cells”. This statement is misleading because it indicates that tumor-associated fibroblasts, immune cells and endothelial cells are also tumor cells.
- Lines 55-57: The authors seem to suggest that tumor cells far away from blood vessels are quiescent and ultimately die. They seem to overlook the fact that these cells change their gene expression profile due to HIF-1alpha and learn to survive under the hypoxic and low-glucose conditions. See the authors’ own statement on lines 66-67.
- Paragraph on lines 73-78. The animal model description is confusing. Is it a chemically induced spontaneous colon cancer model in rat or is it subcutaneous injection of tumor cells in nude mice/rats?
- Lines 107-109. It is not clear what the authors want to say here. Warburg hypothesis does not mention that normal cells cannot produce lactate. Normal cells increase lactate production under hypoxic conditions than under normoxic conditions. In contrast, cancer cells produce massive amounts of lactate even under normoxic conditions.
- Table 1 and Fig. 1 describe the same data. It is just a duplicate representation of the data from the same set of experiments.
- 2 uses the same data in Fig. 1 along with data for normal cells. This is not an acceptable method to prepare manuscripts because the same data are used in two different figures.
- 3. How do we know that the fluorescent signal is specific for MG? It appears that the fluorescent signal is greater in NBF than in NDF. Is this true when MG levels in culture medium is measured? If this is true, it might provide some evidence that the fluorescent signal may be specific for MG. Otherwise, the data in Fig. 3 mean very little. We do not even know if there is any level of non-specific fluorescence signal, and if there is, how much of this signal is in comparison to MG-specific signal.
- 5. Again the data for PCS cells are the duplicate of what is provided in Fig. 4.
- Data in Table 5 are the same as that given in Fig. 6. Same is true for the data in Table 6 and Fig. 7. The same problem with the data in Table 7 and Fig. 8.
- The authors have investigated the production of MG and lactate under four different conditions: normoxia and hypoxia/low and high glucose. They have used a number of tumor cell lines and different normal and transformed non-tumor cells. However, the data do not provide any significant insight into the biological relevance of the data except that the data provide a comparison of MG and lactate production during in vitro culture.
- In summary, the studies reported in this manuscript do not address any particular question related to the role of MG and lactate, produced either by tumor cells or by stromal cells, in tumor growth.
Author Response
Comment 1
The manuscript could benefit from a more careful editing for English language.
Response 1
We have deeply reviewed the editing of the paper for English language.
Comment 2
Abstract, 1st paragraph: how do you get tumors in cancer-free patients? How do you show correlation between circulating levels of MG and tumor growth in cancer-free patients?
Response 2
Indeed the wording mistake in the abstract has been corrected. Thank you
Comment 3
Line 47: “these different categories of tumor cells”. This statement is misleading because it indicates that tumor-associated fibroblasts, immune cells and endothelial cells are also tumor cells.
Response 3
There is a confusion between cancer cells and tumor cells. Tumor cells include cancer cells and stromal cells. If we make this distinction, line 48 is not misleading.
Comment 4
The authors seem to suggest that tumor cells far away from blood vessels are quiescent and ultimately die. They seem to overlook the fact that these cells change their gene expression profile due to HIF-1alpha and learn to survive under the hypoxic and low-glucose conditions. See the authors’ own statement on lines 66-67.
Response 4
Intratumoral cell apoptosis/necrosis is a hallmark of cancer. Here too, we should distinguish between cancer cells from stromal cells. Clearly stromal cells may die because they cannot adapt. This may also partially concern cancer cells, due to lactate acidity (see the discussion section) albeit cancer cells indeed can adapt well enough, as mentioned in line 66-70.
Comment 5
Paragraph on lines 73-78. The animal model description is confusing. Is it a chemically induced spontaneous colon cancer model in rat or is it subcutaneous injection of tumor cells in nude mice/rats?
Response 5
Paragraph on lines 83-88. There is no confusion, the model is a grafted model in rats derived from a chemically-derived tumor. For clarification we added that the model was a tumor grafted model.
Comment 6
Lines 107-109. It is not clear what the authors want to say here. Warburg hypothesis does not mention that normal cells cannot produce lactate. Normal cells increase lactate production under hypoxic conditions than under normoxic conditions. In contrast, cancer cells produce massive amounts of lactate even under normoxic conditions.
Response 6
This is an important point of discussion. If we read again the original Warburg’s papers before he published in 1956 in Sciences his conceptualized paper entitled “On the origin of cancer cells”, he never used cells, but instead tumor and normal tissues. We believe that there is a confusion in the present scientific literature context transposing the Warburg effect evidenced from tumor tissue to isolated cancer cells. Furthermore, Warburg presented the lactate production by tumor tissues to be cancerr specific. To our knowledge he never said that normal cells could also produce lactate in aerobic conditions. This confusion persists to such a degree that the non-specificity of the Warburg effect has been the object of several publications that we mentioned in the discussion, showing that several categories of non-cancer proliferating cells may be characterized by a Warburg effect (see our discussion section 3). This is why our study tends to show that the Warburg effect is not restricted to cancer cells. Please could you provide us original scientific publications showing the Warburg effect is proved to occur in isolated cancer cells and may also be generalized to all categories of normal cells in normoxic conditions, so we could include this in our paper.
Comment 7
Table 1 and Fig. 1 describe the same data. It is just a duplicate representation of the data from the same set of experiments.
Response 7
You are right that Table 1 and Figure 1 describe the same data. We have therefore suppressed all tables from the core manuscript to keep only figures. However all tables, which give raw data have been kept in a supplementary material and methods that we added to the revised manuscript. This is indicated in lines 206 and 207 of the manuscript new version.
Comment 8
2 uses the same data in Fig. 1 along with data for normal cells. This is not an acceptable method to prepare manuscripts because the same data are used in two different figures.
Response 8
In figure 2, we used mean free MG release level in cancer cell lines instead of data reported in figure 1 to compare data from normal cell lines to those of cancer cell lines.
Comment 9
How do we know that the fluorescent signal is specific for MG? It appears that the fluorescent signal is greater in NBF than in NDF. Is this true when MG levels in culture medium is measured? If this is true, it might provide some evidence that the fluorescent signal may be specific for MG. Otherwise, the data in Fig. 3 mean very little. We do not even know if there is any level of non-specific fluorescence signal, and if there is, how much of this signal is in comparison to MG-specific signal.
Response 9
It is not possible to say from the two types of picture that the fluorescent signal is greater in NBF (normal cell lines) than in NDF (normal controls). The two types of cells were cultured in the same conditions (see material and methods). The anti-MG probe was provided by Dr D. Spiegel from the Yale University (USA) and considered to be specific to MG. We have no doubt that the fluorescent staining was specific to MG in both types of culture.
Comment 10
Again the data for PCS cells are the duplicate of what is provided in Fig. 4.
Response 10
We agree. We suppressed all tables from the revised manuscript (see point 7).
Comment 11
Data in Table 5 are the same as that given in Fig. 6. Same is true for the data in Table 6 and Fig. 7. The same problem with the data in Table 7 and Fig. 8.
Response 11
Same comment as in point 7 and 10.
Comment 12
The authors have investigated the production of MG and lactate under four different conditions: normoxia and hypoxia/low and high glucose. They have used a number of tumor cell lines and different normal and transformed non-tumor cells. However, the data do not provide any significant insight into the biological relevance of the data except that the data provide a comparison of MG and lactate production during in vitro culture.
Response 12
We agree that there are some limitations of our study. However we discussed the biological relevance of our study in the discussion section, taking into consideration our previous publications (free MG as a metabolic new marker of tumor progression) and from other publications from the scientific literature. We changed the title which we restricted “implications” of our culture data.
Comment 13
In summary, the studies reported in this manuscript do not address any particular question related to the role of MG and lactate, produced either by tumor cells or by stromal cells, in tumor growth.
Response 13
The objective of our paper was not to prove the role of free MG or lactate in tumor growth (we had proved it previously in specific published work for free MG and it has been shown in many scientific papers for lactate), but to show whether non-cancer cells could also produce free MG and lactate in different micro-environmental conditions (oxygen and glucose) to hypothesize that in tumors stromal cells could do the same. We feel it is clearly indicated in the introduction. To validate the hypothesis that activated stromal CAFs could produce and release free MG, experimental transcriptomic work using coculture studies is ongoing.
Reviewer 3 Report
Comments and Suggestions for Authors
This reviewer finds a very low level of novelty and a poor preparation of the manuscript. Moreover, often the findings not always support the claims of the authors. The resubmission of an in deep revised version is encouraged. Some major issues are evidenced in the following:
- A clear description of the pathway leading to MG synthesis, its effects and biological role is missing in the introduction. Instead, it is used as a discussion and to self-reference author’s work. Please summarize and move to the introduction previous data. The discussion is not the place for a review on MG but shold coincisely discuss author’s findings in this manuscript-
- “From this data, we confirmed that the Warburg effect measured at cellular levels by lactate production in aerobic conditions is not specific to cancer cells; and we proposed that in malignant tumors, both cancer and stromal cells contribute to tumor growth and development.” this concept is present in 100% of updated biochemistry and oncology textbooks. In this paper is reiterated several times as a discovery or a confirmation.
“In this culture study using human cells we confirmed that free MG and lactate are permeable and diffusible glycolytic molecules which are released in the culture medium.” well known.
“Regarding lactate release, this study reveals that cells from cancer cell lines cultured n low or high glucose concentration in normoxic conditions increase their lactate produc-tion and release in the culture medium generally at significantly much higher levels than normal cells do.” obviously.
- “Finally the current data led us to hypothesize how free MG and lactate produced and released by cancer and stromal cells may play a distinct role in tumor growth and progression.” I do not see in this paper any experiment on the role of MG and lactate produced by different cells in an autocrine/paracrine way.
- Figure1 is a part of Figure2, avoid data duplication. In general use graphs and delete tables or move them to supplementary, showing the same data in different figures or both as a graph and a table is not appropriate.
- “This suggests that the majority of cancer cell lines have lost their normal glucose uptake and glycolytic regulation, whereas the two MCF-7 and BXPC3 cancer cell lines have kept a certain capacity of normal regulation for glucose uptake and glycolytic activity.” I do not think those data are sufficient for this statement and the following. Moreover, a clear definition of what is the “normal” glucose regulation is missing: indeed ,different cells are well known to regulate glycolysis differentially and the panel of “normal” cells also feature some cells that do not increase MG production in glucose enriched medium and others that do this. I’m missing the reason behind the claim that cancer cells do not regulate their glycolysis despite experimental results shows that lactate and MG changes with glucose concentration and hypoxia (the latter indicating that they can adapt glycolysis rate to meet ATP and precursors demands).
MINOR POINTS
- “Free Methylglyoxal and Lactate Produced and Released by Cultured Cancer and non-Cancer cells: How they differentially contribute to tumor growth and development.”The title is so long that it appears truncated in the site. The second part poorly reflects the content.
- "... in cancer-free patients methylglyoxal (MG), a side-product of glycolysis, is recovered from tumors ....” this makes no sense.
- “normal human cell lines” is ambiguous “non tumorigenic cell lines” maybe more appropriate.
- “In fact, there is a metabolic heterogeneity in tumors, some cells remaining at a glyco- lytic state, some others using OXPHOS predominantly [48,50]. This is the case for cancer cells. This explains why oxidative cancer cells generate ATP to proliferate, while glycolytic cancer cells still undergo a predominant Warburg effect by producing and releasing high quantities of lactate, which may be incorporated and metabolized by proliferating cancer cells for their energetic needs.” Somehow confused, please consider revision.
- The scheme 657-659 is not visualized correctly on my pc.
Author Response
Comment 1
This reviewer finds a very low level of novelty and a poor preparation of the manuscript. Moreover, often the findings not always support the claims of the authors.
Response 1
We have deeply reviewed the submitted manuscript for English language and for reported experimental data. We disagree with reviewer 3 about the “low level of novelty” of the manuscript. Did reviewer 3 read our two previous published papers on free MG and tumor progression? And did he read extensively papers on free MG to claim that our work is not new?
Comment 2
The resubmission of an in deep revised version is encouraged. Some major issues are evidenced in the following: A clear description of the pathway leading to MG synthesis, its effects and biological role is missing in the introduction. Instead, it is used as a discussion and to self-reference author’s work. Please summarize and move to the introduction previous data. The discussion is not the place for a review on MG but shold coincisely discuss author’s findings in this manuscript-
Response 2
Nevertheless we totally agree with reviewer 3 that the manuscript should be improved.
As reviewer 3 recommended, we summarized and moved previous data on MG synthesis and biological properties in the introduction – see paragraph of lines 76 to 82.
Note that the discussion has been shortened. It is not a review on MG, but a discussion about the interpretation we can make from the experimental data we provided, and a discussion on the biological consequence and hypothesis that we learn from it.
Comment 3
“From this data, we confirmed that the Warburg effect measured at cellular levels by lactate production in aerobic conditions is not specific to cancer cells; and we proposed that in malignant tumors, both cancer and stromal cells contribute to tumor growth and development.” this concept is present in 100% of updated biochemistry and oncology textbooks. In this paper is reiterated several times as a discovery or a confirmation.
“In this culture study using human cells we confirmed that free MG and lactate are permeable and diffusible glycolytic molecules which are released in the culture medium.” well known.
“Regarding lactate release, this study reveals that cells from cancer cell lines cultured n low or high glucose concentration in normoxic conditions increase their lactate production and release in the culture medium generally at significantly much higher levels than normal cells do.” obviously.
Response 3
Concerning the Warburg effect, may I remind reviewer 3 that Warburg in articles published before its seminal conceptualized article in Science entitled “On the origin of cancer cells” used tumoral tissue for comparison with normal tissue instead of isolated cells. Please could Reviewer 3 give us some original scientific publications (not reviews) having shown that the Warburg effect applies to isolated cancer cells and apply not specifically to all categories of normal cells? So we could mention these original publications in our manuscript.
To our knowledge, that cancer cells and normal cells (including non-transformed normal fibroblasts) release free MG in the culture medium has not been previously published using such panel of cancer and normal cells, probably because free MG is characterized by its extreme electrophilic reactivity with all intracellular and extracellular macromolecules.
Comment 4
“Finally the current data led us to hypothesize how free MG and lactate produced and released by cancer and stromal cells may play a distinct role in tumor growth and progression.” I do not see in this paper any experiment on the role of MG and lactate produced by different cells in an autocrine/paracrine way.
Response 4
We agree with reviewer 3 that he could not find in the paper any original finding on the role of free MG and lactate in tumor progression, because for free MG we had previously published it in two papers (references 33 and 34 of the manuscript), and for lactate, because it had been published many times by others (see the discussion). So the objective of the present manuscript was not to prove that free MG or lactate could contribute to tumor growth, but to show whether non-cancer cells could produce and release free MG and lactate in normoxic conditions. We feel this objective is clearly indicated in the introduction. Indeed the finding that normal cells can produce lactate in normoxy, has not been reported in the scientific literature, because normal cells are not commonly supposed to produce D-lactate from the intracellular free MG detoxification process by glyoxalases in normoxic conditions.
Comment 5
Figure1 is a part of Figure2, avoid data duplication. In general use graphs and delete tables or move them to supplementary, showing the same data in different figures or both as a graph and a table is not appropriate.
Response 6
We agree with reviewer 3 that both reported Figures and Tables are not appropriate. So in the revised core manuscript we have suppressed all tables which appear in the supplementary material and methods.
Comment 7
“This suggests that the majority of cancer cell lines have lost their normal glucose uptake and glycolytic regulation, whereas the two MCF-7 and BXPC3 cancer cell lines have kept a certain capacity of normal regulation for glucose uptake and glycolytic activity.” I do not think those nse 6data are sufficient for this statement and the following. Moreover, a clear definition of what is the “normal” glucose regulation is missing: indeed, different cells are well known to regulate glycolysis differentially and the panel of “normal” cells also feature some cells that do not increase MG production in glucose enriched medium and others that do this. I’m missing the reason behind the claim that cancer cells do not regulate their glycolysis despite experimental results shows that lactate and MG changes with glucose concentration and hypoxia (the latter indicating that they can adapt glycolysis rate to meet ATP and precursors demands).
Response 7
As far as glucose regulation is concerned, we agree that the difference we observed between cancer and normal cell lines for MG production in the different glucose medium concentration is associated with some exceptions, depending on the type of lines considered. However, the three set of experiments were reproduced in triplicate and free MG and lactate measurement were done in duplicate; so we believe that the results we obtained were robust enough. However accordingly we have discussed this point in the discussion section. But we disagree with reviewer 3 when he questions the fact that cancer cells have lost their capacity to regulate glucose uptake and glycolysis. This is a basic property of cancer cells (see the introduction lines 71 to 75) which is fully demonstrated to be associated with genetic and/or epigenetic mutations.
MINOR POINTS
“Free Methylglyoxal and Lactate Produced and Released by Cultured Cancer and non-Cancer cells: How they differentially contribute to tumor growth and development.” The title is so long that it appears truncated in the site. The second part poorly reflects the content.
"... in cancer-free patients methylglyoxal (MG), a side-product of glycolysis, is recovered from tumors ....” this makes no sense.
“normal human cell lines” is ambiguous “non tumorigenic cell lines” maybe more appropriate.
“In fact, there is a metabolic heterogeneity in tumors, some cells remaining at a glyco- lytic state, some others using OXPHOS predominantly [48,50]. This is the case for cancer cells. This explains why oxidative cancer cells generate ATP to proliferate, while glycolytic cancer cells still undergo a predominant Warburg effect by producing and releasing high quantities of lactate, which may be incorporated and metabolized by proliferating cancer cells for their energetic needs.” Somehow confused, please consider revision.
The scheme 657-659 is not visualized correctly on my pc.
Responses
We have shortened the second part of the tittle manuscript but kept it because it implicates previous work and part of the discussion.
‘In cancer-free patients” we rectified the wording mistake. Thank you.
We prefer use the term “non-cancer” or “normal“ cell lines because the term “non tumorogenic” may be inappropriate, since normal cell lines in certain in vitro circumstances may be tumorogenic (see one of our very old papers in the journal Cancer in 1974 !).
Round 2
Reviewer 1 Report
Comments and Suggestions for Authors
The author has addressed my concerns
Reviewer 2 Report
Comments and Suggestions for Authors
The study reported in this manuscript monitors the production of methylglyoxal and lactate in normal cells and in cancer cells under different oxygenation and glucose conditions. The data show that both normal cells and cancer cells generate methylglyoxal and lactate and that lactate generated in normal cells under normoxic conditions is derived from methyglyoxal (i.e., D-lactate) rather than from the action of LDH-A (i.e., L-lactate). The authors claim two novel findings from the study. First, normal cells do generate lactate but it comes from methylglyoxal; second, normal cells also make lactate which is in contradiction to Warburg effect which the authors claim that the original studies by Warburg showed lactate production from glucose metabolism only in cancer cells.
Critique
- The claim by the authors that lactate generated in normal cells under normoxic conditions is derived from methylglyoxal has not been validated experimentally. Lactate derived from methylglyoxal is D-lactate whereas lactate derived from pyruvate by LDH-A is L-lactate. It is not clear what stereoisomer of lactate was measured in this particular study. The methodology used to measure lactate does not specify whether or not the technique could differentiate between D- and L- forms of lactate. Without this critical information, the authors' claim that normal cells generate lactate solely from methylglyoxal remains unsubstantiated.
- The authors claim that Warburg showed lactate production under normoxic conditions only in cancer cells. This is not correct. Warburg found normal tissues also generated lactate from glucose metabolism under normal oxygen conditions. He found normal tissues during growth and development generated lactate. He also found that normal retina generated lactate. This has been substantiated in recent years that photoreceptor cells in retina as well as astrocytes in brain generate lactate from glucose metabolism. It is also true in T cells when they proliferate in response to activation. Do the authors mean to suggest that lactate generated in these normal cells actually comes from methylglyoxal? Published reports have already shown that these normal cells express LDH-A, meaning that lactate is generated from pyruvate.
- The authors claim that Warburg never demonstrated glucose-derived lactate production in cancer cells and that he did it only tumor tissues. This is also incorrect. Warburg demonstrated this phenomenon using cancer cells derived from Ehrlich ascites.
- The authors claim that the fluorescent probe they used to detect methylglyoxal in cultured cells is specific because they obtained the probe from the investigators who originally developed it. But the original study by these investigators shows that in cultured cells the fluorescence signal from the probe is substantial even in the absence of externally added methylglyoxal.
- In summary, the present study documents methylglyoxal and lactate in normal and cancer cells under different conditions. The lactate production in normal cells is however not anything new; even Otto Warburg found this phenomenon. What is new is that the authors of the present study claim that the lactate generated in normal cells arise from detoxification of methylglyoxal, but this claim remains unsubstantiated. Taken collectively, the data presented in this manuscript add little to our understanding of glucose metabolism in normal cells versus cancer cells.
Reviewer 3 Report
Comments and Suggestions for Authors
My main concerns remain: novelty is very low and some of the conclusions are not derived by the work but by the literature. However, authors improved the quality of manuscript to a sufficient level not to preclude pubblication.